# The Implication of Wearables and the Factors Affecting Their Usage among Recreationally Active People

**DOI:** 10.3390/ijerph17228532

**Published:** 2020-11-17

**Authors:** Anna Hendker, Malte Jetzke, Eric Eils, Claudia Voelcker-Rehage

**Affiliations:** 1Department of Neuromotor Behavior and Exercise, Institute of Sport and Exercise Sciences, University of Muenster, 48149 Muenster, Germany; anna.hendker@uni-muenster.de (A.H.); eils@uni-muenster.de (E.E.); 2Department of Social Sciences of Sport, Institute of Sport and Exercise Sciences, University of Muenster, 48149 Muenster, Germany; malte.jetzke@uni-muenster.de

**Keywords:** usage, wearable, acceptance, tracking, physical activity

## Abstract

Regular physical activity (PA) is associated with health and well-being. Recent findings show that PA tracking using technological devices can enhance PA behavior. Consumer devices can track many different parameters affecting PA (e.g., number of steps, distance, and heart rate). However, it remains unclear what factors affect the usage of such devices. In this study, we evaluated whether there was a change in usage behavior across the first weeks of usage. Further we investigated whether external factors such as weather and day of the week influence usage behavior. Thirty nine participants received a Fitbit Charge 2 fitness tracker for a nine-week period. All participants were asked to wear the device according to their wishes. The usage time and amount of PA were assessed, and the influencing factors, such as weather conditions and day of the week, were analyzed. The results showed that usage behavior differed largely between individuals and decreased after five weeks of usage. Moreover, the steps per worn hour did not change significantly, indicating a similar amount of activity across the nine-week period when wearing the device. Further influencing factors were the day of the week (the tracker was used less on Sundays) and the temperature (usage time was lower with temperatures >25°). Tracking peoples’ activity might have the potential to evaluate different interventions to increase PA.

## 1. Introduction

Physical inactivity is one of the most important public health problems of the 21st century [1,2]. Regular physical activity (PA) contributes to the prevention of obesity and chronic diseases [2,3]. However, different reasons (e.g., lack of time, feeling too tired [4], or bad weather [5]) might impede people from engaging in a healthy amount of PA, as defined by the American College of Sports Medicine [1]. In recent years, different prevention strategies have sought to overcome these barriers to physical activity. One of these strategies relates to the progressive technologization and affects current tendencies in the fields of sport and prevention. One popular tool to increase PA behavior are wearables, mobile intelligent devices that track one’s steps or heart rate with a wristband.

Wearable devices have gained tremendous momentum and have become part of people’s daily lives by tracking one’s level of activity. In Germany, 6.9 million people used a wearable in 2020 [6]. Since 2016, wearable technology has been ranked in the top three fitness trends [7], showing the growing use of activity trackers. 

Health insurance companies recognize the underlying potential of compelling people to engage in PA by using activity trackers. However, related offers were not always successful, likely to be because they did not correspond to individual preferences or previous experiences. Our aim was to evaluate the usage of such devices in the first weeks after delivery and check possible affecting external parameters.

Previous systematic reviews and meta-analyses indicate that eHealth interventions that implement activity trackers might be an effective tool to promote PA among adults of various ages [8,9,10,11,12]. The general assumption is that once people use a device, they will become concerned with their daily PA, although the positive effects and exact conditions of this usage over the long term are unclear [13,14]. For example, in overweight people, activity monitors were shown to increase the PA level in combination with lifestyle or activity interventions [15]. A cross sectional online survey in a large cohort of mostly young to middle aged adults revealed that the majority of current (81.4%) and former (51.3%) users believed that they incorporated more PA into their days while wearing their activity trackers [15]. Other researchers found increased moderate to vigorous PA and increased steps/day after 16 weeks of activity tracker usage [16]. In this case, wearable technology helped to increase daily PA levels, although this increase seemed to be more effective in individuals with lower baseline PA [17]. Studies on the use of wearables among patients showed different acceptance levels, with about one-third of participants dropping out and only half achieving their daily target goals (e.g., 10,000 steps per day) [5,18] for at least 50% of days [19]. Only about 1/3 of subjects suffering from a myocardial infarct actually used an offered tracker [20]. In the older population, wearable studies led to inconsistent results [19,21]. However, there is no evidence of a positive effect when such interventions are compared to alternative interventions [14], regardless of target group. For example, studies stated no significant difference in PA behavior compared to a control group, at least after six months of wearing an activity tracker [22].

Results on usage behavior over time are also heterogenous. The majority of participants show phases of very consistent activity tracker usage [20,21]. While some are very enthusiastic for months or even years, others lose their interest after some time or do not participate at all. Indeed, the use of activity trackers is often abandoned after a few weeks or months [15,22,23], and only a little over 40% still use such devices after 24 months [23]. Among undergraduate students, 65% stopped using the devices within the first two weeks [24]. To date, little detailed data are available on short-term PA effects (on the first weeks of usage) of wearables.

Moreover, studies have mostly evaluated the summed steps per day [18,25,26,27,28] instead of the steps per worn hour or minute (cadence), i.e., the intensity of PA while wearing the device. The few available studies focused mostly on the usefulness of the measure itself instead of the usage behavior [29,30,31,32]. Attaching a device, however, might not only encourage people to do more steps per day, but also to perform more steps per hour, thereby increasing daily physical intensity or walking speed [19,33]. More research is needed to evaluate the possible changes in steps per hour while wearing such a device.

To sum up, it remains unclear if the usage of wearables leads to an increase in PA since some analyzed cohorts are not representative of others. Furthermore, a possible increase in PA might be accompanied by increased PA over a longer time-period [18,34] and/or by increased physical intensity.

It has also been discussed whether (beside other factors) the day of the week or weather conditions may play a role in human PA patterns. Generally, certain weather conditions might have greater influence on PA during the weekend than during workdays [35]. As intervention studies run over several weeks and thus are likely to cross different seasons, it is important also to integrate weather-dependent activity changes. Diverse studies in different countries seem to show comparable reactions with a lower PA level during colder seasons [36]. Extreme weather conditions with mean temperatures ≥ 29 °C, as well as higher amounts of rain or snow, were shown to lead to a decrease of PA in north America [5,36], especially under severe weather such as rain in summer or ice and snow in winter [37,38]. In Europe, where our investigation took place, lower PA was associated with lower temperatures, heavier rain, and shorter day lengths. Furthermore, lower PA corresponded to temperature cut-offs below 10 °C [39,40]. A decline in PA from November to March is, therefore, expected [41]. Besides weather conditions, typical patterns for each week or day can be observed since usage days are not equally distributed each week [20]. The distribution levels over the week are somewhat contradictory, with both less [20] and more [23] reported usage on Tuesdays and both higher [20] and lower [21] amounts of usage on weekends. Besides usage, more steps could be recorded on the weekend compared to weekdays [20,42]. Jeong et al. reported high average wear times of about 8 h per day at weekends versus 11 h per day on Mondays and Tuesdays [21]. 

### The Current Study

Thus, it remains unclear to what extent wearables are used and affect changes in the PA of new users. With diverse average usage times of just a few weeks up to about seven months in the adult population, more research is needed to understand the requirements for long-lasting use (and effect) of wearable devices such as activity trackers [15,20]. Existing studies focused on different populations (elderly, diseased individuals, or different age groups) and reported heterogenous usage behaviors. Therefore, it is difficult to determine the usage behavior of a recreationally active sample of young to middle-aged adults and what influences usage. 

In this study, we aimed to evaluate how an activity tracker was integrated into daily life and whether there was a change in usage time across a 9-week period since former research indicated a decline in PA during this first phase after the device was introduced [22,23]. The usage time per day and changes after a certain amount of use might be an important indicator for a decline in usage. We hypothesized that interest in a newly implemented wearable device decreases in the first weeks but is highly diverse between individuals. Investigating how fast this process occurs and how it differs between individuals is one aim of this study.

Since about 70% of wearable users reported an initial increase of PA after starting tracking, and 10% of current and 27% of former users reported that their PA levels subsequently decreased to baseline levels within the first weeks of wearing the device [15], a reduction in the amount of PA seems to be expectable. We were further interested in determining the intensity of PA behavior measured as steps per hour when using an activity tracker and if the steps per worn hour change across time. An altered rate of activity per hour might suggest an accompanying change in usage behavior. That is, some people might use such devices only when they are physically active.

Further, affecting parameters of usage behavior need to be better understood. Therefore, we analyzed usage behaviors over the whole week and on different days of the week. We hypothesized that there would be differences in usage behavior and activity level throughout the week. Further, it is not clear how and to what extent environmental factors such as weather conditions influence usage behavior. A natural response might involve reduced PA under “bad” weather situations such as extreme temperatures or a low amount of sunshine per day. We hypothesized reduced PA on extremely hot or cold days and a positive influence of sunshine.

The random transfer of such devices to, e.g., policyholders might increase their usage and, therefore, lead to a conscious and increased use of such devices in everyday activities. Therefore, we ran this investigation on a “normally” active (defined as being physically active at least two hours per week) population that was recruited by chance without specific interest in wearable technology. Besides individual interests and differences, this investigation intended to show what external parameters like weekdays or different weather conditions can affect usage. Existing studies often analyze these factors via questionnaires or retrospective surveys [15,43] among participants that acquired the devices by themselves. In our study, we used the quantitative and physiological data of the tracked subjects. These results will help to assess the acceptance level of wearables and the handler’s usage behavior to identify the influencing factors within the first nine weeks after delivery.

## 2. Methods

### 2.1. Participants

Originally, 81 recreationally active participants between 21 and 47 years of age from two cities in Germany (Münster: 310,000 inhabitants; Cologne: 1.06 million inhabitants) were recruited by notice boards, personal networking, and email distribution through cooperative partners like a health insurance company and different industrial companies. Participants were not specifically informed that they would receive a wearable for this study to avoid impacting the recruitment by chance. 

Subjects were screened by a short telephone interview. Inclusion criteria were an age between 18 and 48 years and being physically active for at least two hours per week. PA was defined as planned physical activity in a non-professional way. Participants were excluded if they owned a wearable, had used one before, or if they reported any history of disease that limited their ability to perform recreational sports or PA in their free time.

Half of the cohort (n = 39) was assigned to the wearable group (25 female, 14 male; age: 32.8 ± 6.9 years, 21–47 years; height: 174.9 ± 10.1 cm; weight: 72.3 ± 14.3 kg) by simple randomization (pulling a match) and received a Fitbit Charge 2 (Fitbit Inc., San Francisco, CA, USA) for a nine-week period. In this paper, we focused only on that cohort. All participants were well educated, i.e., had at least a high school diploma and professional training or course of studies. Recruited participants performed a wide variety of sports (e.g., strength training, team-ball sports, dancing, horseback riding, climbing, rowing, and endurance training like running or cycling). All participants were informed about the design and the procedure of the study and gave their written informed consent. The participants also signed an informed consent to allow us to access to their recorded wearable data. There were no dropouts over time because the usage of the wearable was tracked across the whole study period, and non-usage (without a specific reason, e.g., illness) was also treated as a result. This study was approved by the local Ethics Committee of the Department of Psychology and Sports Sciences, University of Muenster, Germany (2019-33-AH) and was conducted according to the Declaration of Helsinki.

### 2.2. Measures 

Physical activity data and the usage of an activity tracker were evaluated via a conventional device (Fitbit Charge 2; Fitbit Inc., San Francisco, CA, USA). Participants were asked to wear the device on their preferred wrist and to secure data transmission by adjusting the device to a proper position. This wearable device is a wireless, triaxial accelerometer with a heart rate sensor. The raw acceleration signals are converted into sums to estimate a person’s steps. Further, one’s heart rate, activity level and energy expenditure per minute were extracted [44,45]. To secure their personal data, participants received anonymous email addresses to create an account, and help was offered to set up the account if a participant had any problems in doing so. The participants then synchronized the device with their smartphone app for better data visualization and to assure synchronization without connecting the device to a personal computer. Fitbit data synchronization and data securing were performed by the subjects and the scientific staff at least once a week. If a participant did not synchronize in time or did not connect to the webpage at all, he or she received an email or a phone call with help offered. Data were then downloaded anonymously as JSON files by the study staff and processed by the R 4.0.2 base package [46]. Data analysis used the heart rate and steps recorded by the wearable. 

The usage time was determined by the HR readings per minute. The heart rate was recorded constantly, and the data output generated small clusters of reading intervals produced by Fitbit©. Furthermore, Fitbit provides a confidence indicator for measuring parameters to indicate the level of confidence for data correctness. In this study, we only used values that had a confidence > 0. A valid use per minute was assumed if at least one HR value was available in that minute. From these valid uses minutes, the total usage time was calculated for each participant each day. The average usage time was determined as an indicator for acceptance. If there were no HR readings on one day, the usage time is set to 0. The usage hours were analyzed by the hours per day.

For the physical activity analysis, a maximum of 16 h was analyzed to exclude possible sleep time. The physical activity behavior was measured by the number of steps recorded with the triaxial accelerometer of the Fitbit each day. If at least one step per hour was detected, the hour was considered. In 303 cases (out of 2379 cases, n = 39, 61 days), there were no steps recorded although the wearable was used. These cases were set as “missing”. Steps were analyzed as steps per day.

Additionally, the number of steps per use hour were assessed as a measure of intensity. Since participants were not forced to wear the device constantly, we could only verify the parameters of the hours when the wearable was attached. We excluded days where no step was recorded but used the HR to avoid measurement errors. If the wearable was not used at all during that day, the steps per hour were determined as 0. Further indicators of usage behavior were the number of days since the wearable was received (61 days in total), the day of the week, the number of hours of sunshine in the day (M = 6.0, SD = 4.5, 5 categories: 0 h (13.4%), 1–4 h, (30.2%), 5–8 h (22.3%), 9–12 h (23.8%), > 12 h (10.3%)), and the maximum temperature that day (M = 19.5, SD = 6.3, 5 categories: ≤ 10 °C (5.7%), 11–15 °C (25.1%), 16–20 °C (27.3%), 21–25 °C (21.7%), > 25 °C (20.2%)). Weather information was obtained from a weather data platform for that specific region and was freely available with an hourly resolution [47]. The hours of sunshine and temperature were divided in consistent segments of five degree or four hours, respectively. Weather conditions were calculated individually for the nine-week time span for each participant. Since our subjects lived in an area close to the city center of their hometown and made no long journeys or holidays during the study, weather conditions from the city were used for the analysis. For each participant, weather data of their specific hometown (the two relevant cities lying 150 km apart) are used. 

### 2.3. Procedure and Intervention 

The study ran between April and November 2018 to exclude the cold weather phase. During their first visit to the investigation center, the subjects filled out questionnaires regarding their usual weekly PA followed by an assessment of anthropometric data. Then, a Fitbit Charge 2 was handed out with the request to start usage the upcoming weekend. Participants were asked to wear the Fitbit during the following nine weeks as much as they desired. The device must be charged about every five days for a maximum of two hours. This time was not documented since we assumed that only a few subjects wore the device all the time. The participants received no specific information about the purpose of the study but were informed that we were interested in how they would use the activity tracker and how active they are. The default settings of the device aimed for 10,000 steps/day [48]. If this goal was reached, fireworks combined with vibrations appeared on the wearable’s display. The subjects were encouraged (not forced) to adjust this goal or to test the tracking mode of the device’s sleeping states. They were further told that they were free to handle the device but had to synchronize the device at least once a week. After the nine-week period, participants returned the wearable. 

### 2.4. Statistical Analysis

Pre-processing of the data and statistical analysis were performed using the R 4.0.2 base package [49]. The additional modules “lmerTest” [49] and “lme4” [50] were used for statistical modelling. Statistical analyses were constructed in the same way for both dependent variables (usage time (h) and steps/h). Because the subjects were not forced to wear the device, PA was detected only when the wearable was applied. Therefore, only steps per hour instead of steps per day (PA) were evaluated. In the first step, we assessed the need to use multilevel analysis by comparing a baseline linear mixed effect model that includes only the intercept with the same model, allowing the intercepts to vary across participants. Due to the significant difference between these models, we concluded that intercepts varied significantly across participants and thus used random effects in subsequent analyses. To examine the main effects for the factors week of intervention, weekday, hours of sunshine, and temperature, a linear mixed effects model was calculated for both the outcome variables of usage time and steps per hour, using Satterthwaite’s degrees of freedom method to calculate the p-values [49]. For the main effects, F-statistics were reported, and the omega squared (ω^2^) was reported for the effect size. Upon reaching significance (*p* < 0.05), the model contrasts were further inspected. In the second step, for each dependent variable (usage time and steps per hour), a full linear mixed model with all predictors (weeks, weekdays, temperatures, and hours of sunshine) and a random intercept for the participants was conducted.

## 3. Results

### 3.1. Usage Behavior

Descriptive statistics for the dependent variables are shown in Table 1. The distribution of the average usage time (Figure 1) differed between 0 and 16 h and showed that the wearable was used very differently, including both throughout the whole day and not at all. Over the entire study period and among all participants, the device was used on average 13.08 ± 9.3 h/day, with a median of 15.1 h/day. Thirty three percent wore the watch longer than 16 h per day, but three participants hardly ever wore the watch at all. In 74% of all included days, the wearable was worn for more than 2 h. This amount changed only slightly when cut offs of 4 h (71%), 6 h (68%), and 8 h (67%) were applied. A total of 65% valid days were recorded when focusing only on days that the device was worn for at least 10 h per day. Twelve users wore a device for more than 90% of the days, and 16 users for more than 50% of valid days. 

As depicted in Figure 2, there was a decline in usage across the times of intervention. A linear decrease in usage time is indicated from day one to day 61, as shown by the steps and hours worn (cf. Section 3.1.1). The intercepts also significantly varied across participants. Including random intercepts in our mixed linear effects model showed that the intercepts varied significantly between participants χ^2^(1) = 1110.15, *p* < 0.0001. Thus, the following models all include random intercepts for the participants.

#### 3.1.1. Effect of the Study Week on Usage Behavior

When calculating the linear effects model with random intercepts and the week of the intervention as an independent variable for the dependent variable usage behavior, there was a significant main effect observed for the week of the intervention (F(8,2334) = 15.34, *p* < 0.001, ω^2^ = 0.047). Post-hoc comparisons between all nine weeks show significantly lower usage between each of weeks 6, 7, 8, and 9 compared to weeks 1, 2, 3, 4, and 5. 

#### 3.1.2. Effect of Weekday on Usage Behavior

Figure 3 shows the usage behavior over the week. There was a significant main effect of the weekday (F(6,2334) = 4.43, *p* < 0.001, ω^2^ = 0.009), with lower usage on Sunday than on Monday, Wednesday, and Thursday and lower usage on Saturday than on Wednesday (always *p* < 0.05).

#### 3.1.3. Effect of Temperature on Usage Behavior

There was also a significant main effect of temperature (F(4,2339.4) = 6.48, *p* < 0.001, ω^2^ = 0.009), with lower usage time on days with a maximum temperature of more than 25 °C than 21–25 °C and lower usage on cold days with a maximum temperature less than 10 °C compared to days with maximum temperatures of 16–20 °C and 21–25 °C (always *p* < 0.05) [Figure 4].

#### 3.1.4. Effect of Sunshine Hours on Usage Behavior

Figure 5 shows the usage behavior with regard to sunshine. There was a significant main effect of hours of sunshine (F(4,2339.7) = 3.17, *p* = 0.013, ω^2^ = 0.004), with higher usage on days with 1–4 h and 9–12 h of sunshine compared to days with 0 h of sunshine (*p* < 0.05). 

#### 3.1.5. Full Model: Effects of Weeks, Weekdays, Temperatures, and Sunshine Hours on Usage Behavior

Ultimately, a full linear mixed model with all predictors and random intercepts for participants was developed. Since we hypothesized that usage declines over time, the week was included as the first predictor in the final model. “Week” refers to the weeks since the wearable was received. Subsequently, we included the variables of weekday, temperature, and sunshine to control for the week effect. All three expansions of the model were significant when adding the week (χ^2^(1) = 119.87), *p* < 0.0001, the weekday (χ^2^(1) = 33.69, *p* < 0.0001, χ^2^(1) = 33.69), and sunshine and temperature (*p* < 0.0001 and χ^2^(1) = 15.78, *p* = 0.046). The regression parameters of our full model (Table 2) indicate an intercept of 9.6 h. From week 6 until week 9, there were significant estimates between −1.6 h and −2.8 h. Additionally, all days had higher usage than Sunday, with estimates between 0.9 h and 2.2 h, but there was no significant effect of temperature or sunshine hours.

### 3.2. Cadence

Following the same procedure used for usage behavior, we examined the influence of the main factors of week, weekday, hours of sunshine, and temperature on the recorded steps per hour while the device was actually worn. As the subjects were not forced to wear the device, PA was detected only when the wearable was applied. Therefore, only the cadence instead of steps per day was evaluated. For usage behavior, intercepts varied significantly across participants for cadence, χ^2^(1) = 556.54, *p* < 0.0001. Thus, the following models all included random intercepts for the participants.

#### 3.2.1. Effects of Weeks, Weekdays, Temperatures, and Sunshine Hours on Cadence

Linear mixed models showed no significant main effects for weekday (F(6,2031.4) = 0.54, *p* = 0.776, ω^2^ = −0.001), temperature (F(4,2039,8) = 1.92, *p* = 0.105, ω^2^ = 0.002), or sunshine (F(4,2040.8) = 1.07, *p* = 0.370, ω^2^ = 0.000). However, week had a significant effect on steps per hour (F(8,2032.3) = 4.51, *p* < 0.0001, ω^2^ = 0.014), with a lower number of steps per hour on week 9 than any other of weeks 1 to 8. 

#### 3.2.2. Full Model: Effects of Weeks, Weekdays, Temperatures, and Sunshine Hours on Steps per Hour

We hypothesized that the steps per hour would decline over time. We built the final model by adding this predictor. Accordingly, we first added the variable week to our initial model on usage behavior, which defined the week since the wearable was received. Subsequently, we included the variables of weekday, temperature, and sunshine to control for the week effect. In week 9, there was a lower number of steps per hour (−175.86 steps/h) than in week 1. Additionally, on Thursdays, an additional 77.90 steps/h were performed. On days warmer than 25 °C, the steps per hour were 135.85 less than that on cold days of ≤10 °C (Table 2).

## 4. Discussion

This intervention study evaluated the usage of wearables and the possible affecting factors in the first nine weeks after delivery among recreationally active, young to middle aged, randomly assigned participants. As influencing factors, we analyzed external aspects like weather conditions, week of intervention, and weekday. In general, the results confirmed our hypothesis that usage time declined over the nine-week period. While usage behavior remained nearly constant until week five, it significantly declined afterward. Different factors such as the weekday or weather conditions also affected usage behavior, which led to reduced steps on Thursdays and at on days with especially high temperatures. Furthermore, decreased usage time was measured on Sundays and under extremely high or low temperatures. These results are not consistent with those of different research groups [20,51].

### 4.1. Usage Behavior

Our results revealed that most participants readily accepted using the activity tracker and that there was basic interest in using such a device among our randomly assigned participants. Nevertheless, we found large heterogeneity between individuals. Individual usage times differed immensely between subjects. Some participants were especially receptive to using the device (with a wearing time for all nine weeks), whereas others lost interest very quickly or hardly ever used the device. We recruited our subjects without telling them explicitly that a wearable would be provided. Thus, we can assume that our participants included not just “wearable-responder” individuals but a general sample of the population. Additionally, we excluded participants who had used a wearable beforehand to ensure that the device was a new tool for everyone. Thus, our cohort should represent the “normal” diversity in the population with regard to interest in wearables. With an overall usage time of 13 h/day, we were able to assess the behavior of our cohort and detect differences between the subjects. Our results for average usage time (13.08 h/day) are in line with other findings, e.g., 10.5 h/day [21]. 

The relatively small differences related to cut offs in hours of use indicate that the devices were often worn either for the whole day or not at all. Twenty-eight out of 39 included subjects had used the device for than 50% of valid days, indicating good general acceptance and interest in using the device, especially during the first weeks. A similar outcome was detected by Auerswald et al. [52], who tracked PA and usage behavior among nursing home residents and found a high willingness to wear a fitness tracker, especially in the first five weeks of application. Similar to our results, the interest seemed to decrease after this time period. Over time, people wore the device less, but some still attached the device before being physically active. Additionally, our findings are in line with former research indicating a comparable decline in usage after just a few weeks in multiple (and younger) age groups [15,22,23]. This might indicate the presence of a preliminary motivational boost directly after the device was handed out and when it was used for the first time. We can only speculate based on personal communications, but in a short interview at the end of the investigation some subjects reported being especially receptive to features like the fireworks when hitting 10,000 steps/day or the implemented reminder to move after a certain period of immobilization (features that our wearable provided). The effects of these incentives, however, seemed to decrease after a few weeks, but with a high inter-individual difference. Compared to undergraduate students [24], among whom 75% stopped wearing a fitness tracker after four weeks of use, the young to middle-aged cohort in this study maintained its interest slightly longer. Unsurprisingly, the number of steps and usage time followed a similar negative trend over the period (Figure 2). This indicates that both steps [20,53] and heart rate detection were useful for capturing the use frequency of this device. 

In contrast to our results, nursing home residents significantly increased their average number of steps/day from the first to the fifth week of wearing [52]. In our cohort, significantly fewer steps were performed over the nine-week period. Study outcomes are commonly different, particularly in research with older subjects. On the one hand, studies on people >50 years showed that wearables can be an effective tool to increase people’s PA levels [10]. On the other hand, no increase in PA was measured in the population >65 years compared to the control group after a web-based (plus activity) tracker intervention [54]. Overall, studies with subjects in different age groups revealed the potential of mobile devices (and seem to use similar approaches to those used for wearables) to promote PA [30], but concrete findings for wearables are missing for age groups comparable to ours.

### 4.2. Cadence

Besides the pure usage and application of the activity tracker, we were interested in the amount of activity tracked during the nine-week period, irrespective of wearing behavior. For this purpose, we calculated the steps per wearing hour as a measure of cadence. As we did not force our subjects to wear the device to avoid influencing their “natural” interest, we could only record the activity when the wearable was worn. Consequently, we could not evaluate the true daily amount of PA but only the PA when the device was utilized. Although, as described above, the hours of use and the detected steps per day decreased over the nine-week period, the steps per hour remained stable or even increased. The amount of PA during the wearing periods did not decrease over the nine weeks of wearing. On weekends, the small gap between worn hours and increased numbers of steps indicates that the participants were slightly more active when attaching the device.

### 4.3. Influencing Factors

The recorded steps were closely related to the hours worn. This tendency differs slightly when we consider the day of week. Although no significant difference could be detected compared to weekdays, a slight imbalance between usage time and steps can be seen in Figure 3. Therefore, we can conclude that the randomly assigned subjects did have high interest in wearing such a device during the beginning of the wearing period; this interest decreased over time, but hardly affected the physical activity level.

### 4.4. Weekday

Additionally, different external factors might also affect usage behavior. Besides the reduced usage in weeks six to nine compared to the first weeks, we were able to show that the day of the week played a role in wearable usage. Our findings indicate significantly lower usage behavior on Sundays than on all other days. Other research also states that the usage days are not equally distributed each week [20]. In contrast to our findings, Meyer et al. [20] reported reduced usage on Tuesdays and increased usage on Saturdays and Sundays. Increased usage on weekends was also reported by Jeong et al. [21]; however, in contrast to Meyer, the higher wearing hours on Mondays and Tuesdays in Jeong et al. compared to the other days of the week are more in line with our findings. One reason for this reduced usage on weekends (beside the day of the week) might be that the device did not fit current fashion trends and thus was not applied on the weekends. For example, some subjects complained about the sporty and colorful design. Unfortunately, we did not assess these influencing factors, but they should be included in future studies. This might also explain why this device was worn less at very high temperatures. Here, the relatively wide plastic wristband might also have disturbed some subjects. 

### 4.5. Weather

When considering the influence of different weather conditions on people’s PA behavior, we unsurprisingly found that temperature affected the application. Extremely high or low temperatures led to less usage and, therefore, fewer detected steps per day. Again, this might be due to several reasons. The sporty and wide plastic wristband might have been less comfortable for the subjects and led to reduced applications. Weather conditions with a few hours of sunshine and medium temperatures seemed to yield more wearing hours compared to days with no sunshine at all. In our study, higher usage under 1–4 h and 9–12 h of sunshine was detected (see Figure 5). Former research [36] found that extremely high temperatures lead to reduced PA [51]. In accordance with European studies [40,55], significant effects of temperature were detected in the present research with lower PA at lower temperatures. Like other international studies, we found a decrease in PA at high temperatures >25° C. This finding conflicts with Klenk et al. [40] but is in line with Wu et al. [55]. This difference could be explained by the different survey periods considered in these studies (our intervention took place between April and October to overcome reportedly decreased PA between November and March [41]) and because extremely low temperatures were not recorded. Other research with activity tracker applications reported constant usage throughout the year [20]. Again, the device might not have been worn on especially hot days, at least not when the participants were active, and certain weather conditions had a greater influence on PA behavior during the weekend than during workdays. This effect was not found for all subjects, indicating that such interrelationships should be investigated on a single subject level [35]. More research is needed to distinguish between weather-dependent activity changes over the year. 

One strength of our study is concurrently a limitation: we did not force the participants to wear the device continuously. Therefore, we were only able to verify PA behavior when the participants wore the device, which made it difficult to judge real PA changes over the weeks. We, therefore, assume that subjects used the wearable only during certain hours of the day, especially when active, e.g., during training sessions, or particularly inactive, such as sitting in an office. Our results on intensity measured as steps per hour suggest that the activity tracker was most commonly worn when participants were physically active. To ensure that we included only the hours awake, we decided to use a cut off with a maximum of 16 h when we calculated the cadence. This might have canceled out some possible active time slots but was necessary to obtain less doubtful and more realistic activity values. In addition, we did not check for charging times and if a potential usage time was lost due to a low battery. As a further limitation, measurement errors might have occurred when participants rode a bike over uneven surfaces or, as in our case, engaged in horseback riding. Nevertheless, multiple Fitbit devices were validated [56,57,58,59] and shown to be useful and valid tools to track activity correctly. 

## 5. Conclusions

Our results show that wearables are an interesting tool for young to middle-aged adults, even when individuals are not especially interested in these technical devices. In our study, there was high variability between single subjects in general usage time and in usage behavior over weeks. From week five until week nine (the end of the intervention) the overall reduced usage behavior remained stable. External factors like temperature affected usage behavior, whereas sunshine did not impact this trend significantly. Lower usage was detected on Sundays and Saturdays compared to the weekdays. Overall, the non-significant main effects of the week, weekday, temperature, and sunshine indicate that the most prominent reason for (non-)use is the individual person user. Given the increasing use of wearables to track one’s own PA and increase active behavior, there is a need for further investigations to gain a deeper understanding of factors that might help individuals use these high-potential devices.

## Figures and Tables

**Figure 1 ijerph-17-08532-f001:**
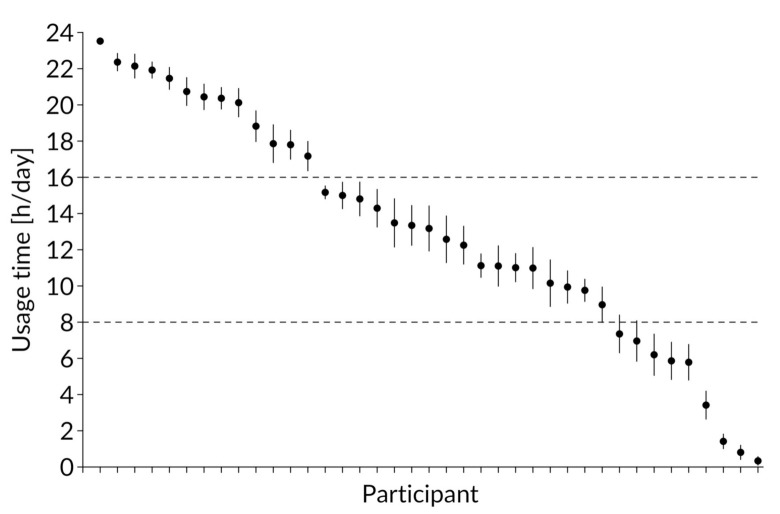
Average usage behavior in the 9-week period for all users.

**Figure 2 ijerph-17-08532-f002:**
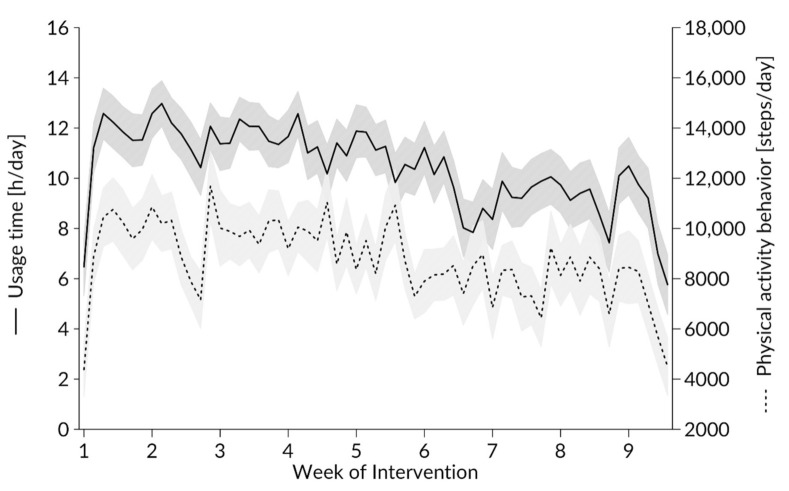
Average use per day of the intervention depicted as usage time (h/day) on the y-axis on the left-hand side and physical activity behavior (steps/day) on the y-axis on the right-hand side.

**Figure 3 ijerph-17-08532-f003:**
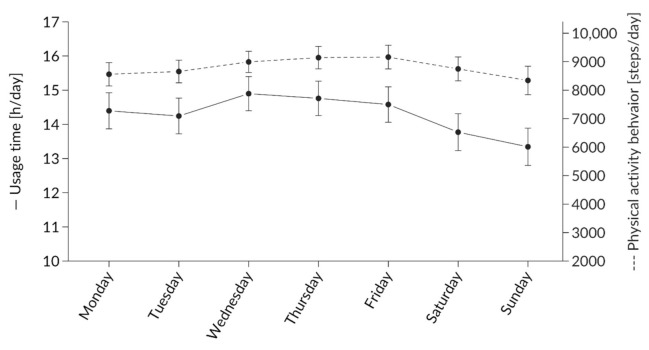
Average use per day of the week.

**Figure 4 ijerph-17-08532-f004:**
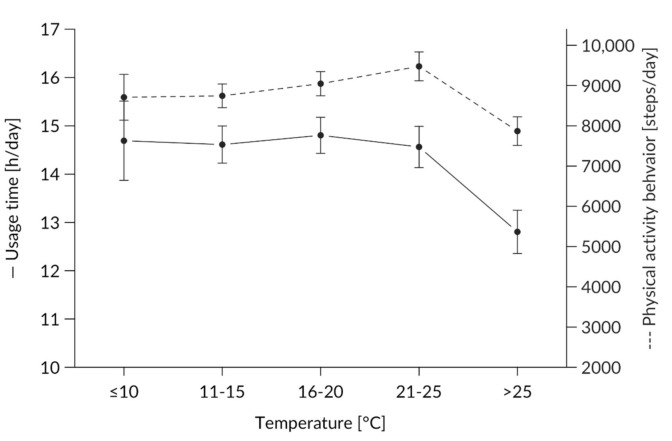
Average use per temperature depicted for usage time (h/day) on the y-axis on the left-hand side and physical activity behavior (steps/day) on the y-axis on the right-hand side.

**Figure 5 ijerph-17-08532-f005:**
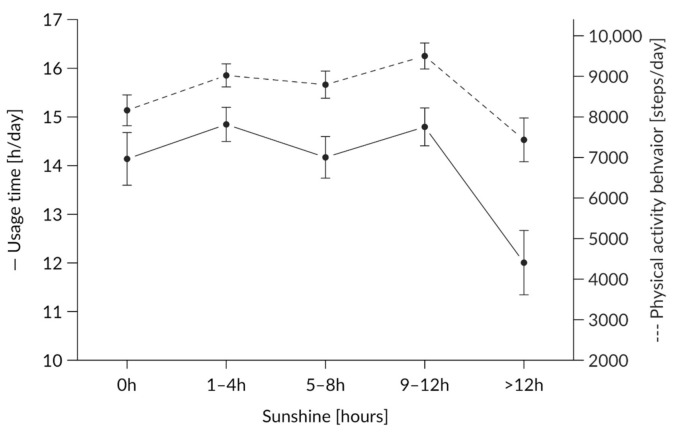
Average use per hour of sunshine depicted for usage time (h/day) on the y-axis on the left-hand side and physical activity behavior (steps/day) on the y-axis on the right-hand side.

**Table 1 ijerph-17-08532-t001:** Descriptive statistics for the dependent variables. (SD = standard deviation; SE = standard error).

Factors		Usage Time (h/Day)					Physical Activity Behavior (Steps/Day)					Cadence (Steps/h)				
		N	Mean	Median	SD	SE	N	Mean	Median	SD	SE	N	Mean	Median	SD	SE
Total		2373	10.45	15.10	6.82	0.14	2070	8813.9	8635.5	7119.7	156.5	2070	642.4	620.2	548.3	12.1
Week	1	273	11.05	16.00	6.71	0.41	259	9187.9	9061.0	7443.8	462.5	259	632.2	625.7	504.6	31.4
	2	273	11.88	16.00	6.28	0.38	255	9579.6	9468.0	7118.0	445.7	255	665.7	648.1	502.7	31.5
	3	273	11.72	16.00	5.98	0.36	253	9930.9	9357.0	7177.0	451.9	253	739.1	662.1	552.4	34.7
	4	273	11.28	15.40	6.39	0.39	239	9708.7	9887.0	6847.7	442.9	239	693.5	690.8	476.1	30.8
	5	273	10.98	14.70	6.57	0.40	231	8976.3	8744.0	6527.0	429.4	231	631.0	621.7	427.3	28.1
	6	273	9.50	12.83	6.99	0.42	225	8238.4	8137.0	6946.6	463.1	225	617.5	602.2	521.5	34.8
	7	273	9.47	13.43	7.06	0.43	228	7715.0	7230.0	6964.1	461.2	228	641.1	572.4	805.4	53.3
	8	273	9.12	12.43	7.23	0.44	225	8183.8	7171.0	7531.8	502.1	225	610.0	564.4	555.0	37.0
	9	273	8.41	11.12	7.47	0.54	155	6850.3	5750.0	6966.4	559.6	155	487.2	433.6	475.6	38.2
Day	Monday	326	10.66	12.47	6.76	0.39	285	8558.4	8199.0	6907.3	409.2	285	638.6	591.0	520.1	30.8
	Tuesday	339	10.51	15.37	6.82	0.37	293	8655.3	8724.0	6822.8	398.6	293	631.6	620.8	519.3	30.3
	Wednesday	351	11.14	15.55	6.63	0.37	319	8992.5	8923.0	6639.2	371.7	319	646.8	630.6	542.7	30.4
	Thursday	351	10.85	16.00	6.69	0.35	315	9142.5	8933.0	6993.1	394.0	315	673.2	629.1	527.2	29.7
	Friday	351	10.51	15.48	6.90	0.36	301	9161.2	9036.0	7221.9	416.3	301	626.1	622.0	473.1	27.3
	Saturday	334	9.97	15.50	6.95	0.37	290	8746.3	8812.0	7144.5	419.5	290	656.2	619.0	642.0	37.7
	Sunday	321	9.40	14.23	6.89	0.38	267	8341.4	6914.0	8179.7	500.6	267	620.6	550.6	609.0	37.3
Temperature (°C)	≤10	134	9.90	13.09	6.76	0.58	111	8708.4	8537.0	5992.0	568.7	111	699.8	668.6	508.5	48.3
	10–15	596	10.56	15.41	6.79	0.28	542	8744.5	8565.5	6847.0	294.1	542	667.6	622.7	636.2	27.3
	16–20	647	10.94	15.85	6.66	0.26	577	9046.6	8933.0	7212.0	300.2	577	641.2	629.1	499.3	20.8
	21–25	516	10.77	15.31	6.73	0.30	441	9477.0	9241.0	7517.0	358.0	441	656.5	626.4	523.5	24.9
	>25	480	9.45	12.83	7.08	0.32	399	7867.9	7331.0	7117.0	356.3	399	578.5	534.8	521.0	26.1
Sunshine (hours)	0	318	10.11	14.28	6.92	0.39	281	8163.0	8116.0	6354.4	379.1	281	641.5	566.7	566.7	33.8
	1–4	716	10.83	15.74	6.72	0.25	639	9023.6	8578.0	7196.5	284.7	639	651.6	537.1	537.1	21.3
	5–8	530	10.27	14.69	6.89	0.30	459	8794.4	8825.0	7174.2	334.9	459	638.0	586.0	586.0	27.4
	9–12	564	10.89	15.39	6.61	0.28	489	9501.8	9483.0	1067.0	319.6	489	683.6	514.9	514.9	23.3
	>12	245	9.12	12.37	7.15	0.46	202	7434.7	6279.5	7660.8	539.6	202	525.5	536.3	536.3	37.7

**Table 2 ijerph-17-08532-t002:** Determinants of usage behavior and steps per hour. Mixed linear models, *p*-values for *p* < 0.05 are shown in bold. (CI= confidence interval; id= participants).

		Usage Behavior (hours)			Steps per hour		
Predictors		Estimates	CI	*p*	Estimates	CI	*p*
(Intercept)		9.56	7.68–11.45	**<0.001**	623.02	467.25–778.79	**<0.001**
Week	1st	Reference			Reference		
	2nd	0.77	−0.09–1.64	0.078	13.05	−68.00–94.11	0.752
	3rd	0.56	−0.31–1.43	0.208	78.21	−3.62–160.04	0.061
	4th	0.18	−0.67–1.03	0.678	51.86	−29.34–133.05	0.211
	5th	−0.03	−0.88–0.83	0.950	−1.90	−84.41–80.60	0.964
	6th	−1.62	−2.49–−0.76	**<0.001**	−36.62	−121.05–47.80	0.395
	7th	−1.61	−2.50–−0.71	**<0.001**	−0.30	−87.72–87.12	0.995
	8th	−1.99	−2.85–−1.13	**<0.001**	−43.51	−127.00–39.97	0.307
	9th	−2.84	−3.79–−1.89	**<0.001**	−175.86	−269.86–−81.86	**<0.001**
Weekday	Sunday	Reference			Reference		
	Monday	1.60	0.80–2.40	**<0.001**	45.75	−33.51–125.00	0.258
	Tuesday	1.57	0.78–2.36	**<0.001**	41.71	−36.59–120.00	0.296)
	Wednesday	2.20	1.41–2.98	**<0.001**	62.66	−14.14–139.45	0.110
	Thursday	1.64	0.86–2.42	**<0.001**	77.90	1.11–154.69	**0.047**
	Friday	1.38	0.60–2.17	**0.001**	23.43	−54.20–101.06	0.554
	Saturday	0.94	0.15–1.74	**0.020**	49.37	−29.59–128.34	0.220
Temperature (°C)	≤10	Reference			Reference		
	11–15	0.32	−0.68–1.31	0.531	−30.06	−129.56–69.44	0.554
	16–20	0.24	−0.85–1.32	0.667	−78.67	−187.19–29.84	0.155
	21–25	0.96	−0.22–2.14	0.112	−50.17	−167.92–67.58	0.404
	>25	−0.30	−1.51–0.91	0.626	−135.85	−256.04–−15.67	**0.027**
Sunshine (hours)	0	Reference			Reference		
	1–4	0.09	−0.66–0.85	0.808	32.01	−42.01–106.02	0.397
	5–8	−0.41	−1.28–0.47	0.363	37.32	−48.42–123.06	0.394
	9–12	−0.21	−1.15–0.74	0.669	67.96	−23.93–159.84	0.147
	>12	0.10	−1.00–1.20	0.855	48.69	−59.98–157.35	0.380
	**Random Effects**						
	σ^2^	25.44			211,520.76		
	τ_00_	19.19 _id_			83,236.19 _id_		
	ICC	0.43			0.28		
	N	39 _id_			39 _id_		
	Observations	2373			2070		
	Marginal R^2^/Conditional R^2^	0.041/0.453			0.18/0.295		

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
