# Peer review of "The Implication of Wearables and the Factors Affecting Their Usage among Recreationally Active People"

_ijerph, 2020, doi:10.3390/ijerph17228532_

Round 1

Reviewer 1 Report

In this study, the authors tried to investigate the factors in relation to the wear time of wearable PA devices. I would like to acknowledge points of interest should be praised; however, there are several concerns that should be improved as follows:

Major:

  1. [Abstract][Introduction] Please explain more simply the background of the Abstract and Introduction section. Also, describe clearly the aim of the study in both sections. The background is too long for explaining the focus of this study and the problem of background.
  2. [Methods] How did the authors deal with the time of charging of Fitbit device for wear time? Was it not a problem for the analyses of wear time? Some participants likely wore the device for over 22 hours, they did not charge the battery?
  3. [Methods] How did the authors get the location information of participants (Fitbit) for matching the weather information.
  4. [Methods] How did the authors divide the categories of the number of hours of sunshine and temperature? What criteria did the authors use?

Minor comments:

  1. Probably use ‘step counts’ instead of intensity throughout the manuscript because the term of intensity usually uses another mean in the PA field.
  2. Please refer to “Table 2” in the body of the manuscript. “Table 1” indicating in 3.1.5 and 3.2.2 might be wrong.
  3. This is my curiously, did consider to add precipitation(rainfall information)for one of weather information? If possible, I would like to see it.

Author Response

Thank you very much for your valuable and helpful comments. We did our best to straigthen the Abstract and Introduction to make the background and the aim of our study more clear. All other changes are marked in the corrected manuscript. We marked the inserted parts in green writing.

Additionally, our manuscript underwent a professional English editing, so some parts are corrected and spelling and grammar was controlled. You will find all these changes in the manuscript as well, but not listed seperately to not impact your clear overview.

The responses to your concrete comments are listet in the PDF file below.

PLEASE SEE ATTACHMENT!

Reviewer 2 Report

Dear International Journal of Environmental Research and Public Health,

thank you very much for the possibility to serve as a Reviewer in a prestigious periodical like International Journal of Environmental Research and Public Health.

About this paper titled “Implication of wearables and affecting factors for usage in recreationally active people”, the contents and the rhetoric by which it was handled are appreciable and the paper is very well organized.

Brief Overview:

The objective of this study was to assess how an activity tracker could be integrated in daily life and if this kind of device could influence physical activity behavior.

The sample was composed by 39 participants. They received a fitness tracker for nine week and accepted to be monitored for this period.

General comment:

This paper examining interesting topics like the influence of an activity tracker on regular physical activity.

Although, the matter of the study is original, minor questions require clarification in order to improve the quality of the manuscript. I have listed below a specific comment to the authors.

First of all, two general advise:

- change the word “persons” with “participants”.

- reduce the introduction, going strictly to the point.

The current study:

- (line 119) Why authors choose nine weeks? There are other studies that used this time?

- (line 129) Why a reduction of the amount of PA is expectable?

- (line144) Which criteria described normal active population?

Methods:

- (lines 160 – 164) Which criteria described normal active population? I suggest to describe exclusion criteria.

Measures:

- (line 182) I suggest to better describe the sensor application (ex. Right or left wrist is the same? Etc.)

Conclusions:

- (line 460) In the results authors affirmed that temperature influenced the results, but in the conclusion, they stated the opposite declaration. I suggest to clarify this point.

Author Response

Thank you very much for your valuable and helpful comments. We did our best to straigthen the Abstract and Introduction to make the background and the aim of our study more clear. All other changes are marked in the corrected manuscript. We marked the inserted parts in green writing.

Additionally, our manuscript underwent a professional English editing, so some parts are corrected and spelling and grammar was controlled. You will find all these changes in the manuscript as well, but not listed separately to not impact your clear overview.

The responses to your concrete comments are listet in the PDF file below.

PLEASE SEE THE ATTACHMENT!

Round 2

Reviewer 1 Report

Thank you for your work in revising your paper.